# Modelling the Combined Effect of Green Leadership and Human Resource Management in Moving to Green Supply Chain Performance Enhancement in Saudi Arabia

Benameur Dahinine [1], Abderrazak Laghouag [2,*], Wassila Bensahel [1], Majed Alsolami [1] and Tarek Guendouz [1]

[1] Department of Management, College of Business Administration, University of Tabuk, Abha 62526, Saudi Arabia; bdahinine@ut.edu.sa (B.D.); wbensasel@ut.edu.sa (W.B.); malsolamy@ut.edu.sa (M.A.); tguendouz@ut.edu.sa (T.G.)

[2] Department of Business Administration, College of Business, King Khalid University, Abha 62552, Saudi Arabia

\* Correspondence: alaghouag@kku.edu.sa

**Abstract:** Previous research has been limited in examining the causal relationship between green transformational leadership (GTL) and green supply chain management (GSCM), with the intermediary influence of green human resource management (GHRM), within the pharmaceutical sector of the Kingdom of Saudi Arabia (KSA). This gap persisted despite the recognition in Saudi Vision 2030 of logistics, specifically, supply chain management (SCM), as fundamental to the national development agenda, given that contemporary competitiveness lies in the efficacy of supply chains (SCs) rather than individual companies. Moreover, the achievement of economic progress hinges significantly on how well these accomplishments align with sustainability demands and obstacles. This paper aims to investigate the extent to which GTL fosters GRHM practices to enhance the maturity of GSCM performance in the pharmaceutical industry in the KSA. In other words, the research goal is to explain the variance of GSCM due to GHRM and GTL. Drawing upon the Resource-Based View (RBV) and the Ability–Motivation–Opportunity theory (AMO), GTL can enhance many aspects of GHRM, such as employee performance measurement, training content design, recruitment criteria, and green-based rewards policies, which positively influence GSCM practices. The methodology employed is deductive and translated into a questionnaire derived from a comprehensive review of the existing literature. This questionnaire was subsequently tested through the collection of 111 responses from pharmaceutical companies operating in the KSA. The results show the critical effects of GTL and GHRM on GSCM in this sector. The research provides fresh theoretical perspectives and actionable recommendations based on the outcomes. As for originality, this research explores the contribution of transformational leadership and green human resource management in enhancing SC sustainability. The research provides directions for future research to investigate the mediating or moderating impacts of other significant factors, such as green thinking or eco-friendly behaviour, on SCM sustainability. As for practical implications, this research came up with an in-depth understanding of SC managers and professionals regarding their practices related to sustainability.

**Keywords:** green transformational leadership; green human resource management; green supply chain management; RBV theory; AMO theory; Saudi Arabia

## 1. Introduction

Saudi Vision 2030 articulates the need for building all capabilities that enhance Saudi supply chain (SC) efficiency and the agility to facilitate trade and improve economic development. The Kingdom aims to establish world-class logistics hubs with high-quality service to ensure seamless logistical processes, increase productivity, and attract the attention of well-reputed companies worldwide. The second agenda of Saudi Vision

consists of investing in human capital and unskilled professionals in SC at all hierarchical levels (decisional and operational levels) through effective collaboration with specialised organizations and research institutions. In general, it is evident that Saudi Vision 2030, in its journey to change the country's economy and make it a worldwide actor in the industrial and logistical sectors, depends absolutely on SC (Saudi Vision 2030).

Supply chain management (SCM) is a business concept that, despite its apparent simplicity, is quite difficult because of the nature and variety of the decisions it involves [1]. Consequently, the definition of this concept and its encompassing elements have evolved in both the scholarly literature and professional consulting organisations [2]. SCM refers to the planning and control of three types of flows: financial, informational, and material. This entails managing and coordinating all SC activities [3–6]. The SC deals with specific issues such as the network structure, the distribution of power and value creation among all SC actors, and the adaptation of the structure due to the change of contingency factors. The SC is interested in global performance instead of individual performance, which means managing the whole process, from the primary supplier of raw materials to the final customer, based on a win–win approach, knowing that SC actors are autonomous and have complete decision-making authority. The SCs organise the collaboration among all the actors through the different levels involved, namely, the strategic, tactical, and operational levels, to meet the final customer's requirements while maintaining a higher level of responsiveness and flexibility [7].

The aspects and characteristics of SCM change with regard to the activities of each sector. For the pharmaceutical industry, tracking systems for drugs and production processes and quality control are highly important. Like any other SC, the pharmaceutical SC deals with producing, transporting, and consuming drugs. The pharmaceutical supply chain (PSC) is an exceptional SC in which drugs are produced, transported, and consumed [8]. The pharmaceutical industry's activities, which comprise medicine development, manufacturing, and distribution, are vital enablers for any healthcare structure. The overarching structure of the contemporary pharmaceutical sector commences with the discoveries of drugs. It advances through preclinical and clinical trials, pharmaceutical manufacturing, and market entry via scientific research and development (R & D) [9–11]. The PSC is of the highest importance due to many facts: (1) The PSC presents a driver for competitiveness and SC surplus, and businesses cannot be competitive if they do not consider and adopt SCM orientation and logic [12]. (2) The PSC tends to make affordable medicine available, especially in underdeveloped nations, where drug prices have been observed to multiply by more than six times beyond the permitted global threshold. Furthermore, the availability of cost-effective medications in the market is restricted. Lack of funding, inaccurate forecasting, absence of incentives for keeping inventories, ineffective distribution methods, and theft of drugs for private resale have collectively contributed to the shortage of essential and affordable medications vital for the primary healthcare sector [13]. Duarte et al. [14] confirm that ensuring the sustained accessibility and availability of pharmaceutical products is one of the main challenges facing pharmaceutical SCs today. (3) A resilient PSC strengthens readiness for emerging outbreaks and crises and could ensure national drug security. Due to the pandemic quarantine, access to medications was hampered globally, endangering the lives of many people. The PSC and the healthcare system face significant challenges due to elevated rates of drug consumption and inadequate local pharmaceutical production [11,15,16]. Pharmaceutical companies also operate in both local and global markets, where they must comply with rules and regulations pertaining to the research, examination, sales, and distribution of pharmaceuticals [17].

Several previous research findings suggest that the primary factors behind medication shortages in Saudi Arabia are associated with deficiencies in supply chain management. Furthermore, there is a lack of regulations mandating timely notification of shortages in pharmaceutical drugs, and the policies related to governmental procurement do not adapt swiftly to shifts in the pharmaceutical market. Additionally, insufficient profit margins for certain critical medications and inefficient legal actions taken against pharmaceutical

companies and authorised importers and distributors, as well as excessive governmental regulations within the pharmaceutical sector, all contribute to this issue [11,18].

Along with the difficulties mentioned above concerning SC activities and processes, the environmental issues that govern the pharmaceutical industry should not be ignored [18]. There is an increasing requirement for businesses and research entities to collaborate on sustainable practices because of expanding regulations, resource constraints, and scientific and public opinion shifts. It is very important in the pharmaceutical industry to make sure that the upcoming generation of scientists possesses the information and abilities required to address and consider sustainability concerns [19]. Human resources is one of the most determinant resources in ensuring the meeting of evolving environmental and ecological requirements. GHRM focuses on integrating environmental concerns into HRM practices. It aims to promote employees' green behaviour in the context of an effective environmental and social responsibility strategy. GHRM encourages staff to engage in sustainable activities and develop environmentally friendly concepts. GHRM practices refer to environmentally conscious hiring and selection, environmental learning, green performance appraisal, remuneration, and engagement [18,20]. GHRM alone, however, is insufficient to provide oneself with a long-term competitive advantage. Other organisational elements, including senior management support and leadership, along with individual characteristics, such as an employee's environmental perspective, can also affect an employee's capacity, drive, and chance to embrace green techniques [18,21,22].

After the discussion above, it appears crucial to assess the level of progress and the maturity of environmentally friendly operations in the pharmaceutical supply chain in Saudi Arabia. This is particularly significant as the pharmaceutical industry plays a crucial role in safeguarding the country's healthcare system and enhancing its readiness to handle any potential epidemics, beyond the ongoing COVID-19 crisis [11]. To achieve this goal, the National Industrial Development and Logistics Program (NIDLP) set the following initiatives: (1) encourage the local manufacturing of generic medications, (2) enhance domestic production of pharmaceuticals based on biology, genetics, and sophisticated therapies. (3) create an innovative technical surveillance system for imported goods. The identified research gap lies in the lack of empirical studies specifically examining the impact of green transformational leadership on green supply chain management (SCM) in Saudi Arabia, considering the mediating role of green human resource management (GHRM). While there might be research on each component individually, there seems to be a dearth of comprehensive studies exploring the interconnectedness and potential mediation effects between these three elements within the Saudi Arabian context. This research gap suggests a need for empirical investigations that delve into how green transformational leadership influences green HRM practices, which in turn affect green SCM outcomes in the Saudi Arabian business context.

Saudi Arabia's unique context, shaped by its heavy reliance on oil exports and initiatives like Vision 2030, significantly impacts the study and implementation of green leadership, HRM, and SCM. Factors such as the regulatory framework, investment in renewable energy, and water scarcity contribute to specific challenges and opportunities in promoting sustainability within the country. As a result of the above discussion, the research question might be crystallised as follows: to what extent does green transformational leadership foster green human resource management practices to enhance the maturity of GSCM performance in the pharmaceutical industry in the KSA? Several theoretical frameworks are proposed in order to develop a research model and hypotheses to address the last research question. The research approach is then described. The validity and reliability of the data is evaluated, and hypothesis testing is performed. Finally, a discussion of the results is given. This study offers several practical implications, as follows: The primary benefit is in providing executives and managers in Saudi Arabia a thorough understanding of the current state of GHRM and GTL practices geared to leveraging GSCM. Finally, this paper concludes by presenting the first study examining the causal relationship between GTL, GHRM, and GSCM maturity in the pharmaceutical sector in Saudi Arabia.

## 2. Theoretical Background and Research Development

### 2.1. The Resource-Based View and the Ability–Motivation–Opportunity (AMO) Theory

For this research, the hypotheses have been developed by merging two principal theories in the field, namely the Resource-Based View (RBV) theory proposed by Barney [23] and the Ability–Motivation–Opportunity (AMO) theory initiated by Bailey [24], as described by Marin-Garcia and Tomas [25], to test how leadership and human resources practices might be a vital resource for leveraging GSC practices and influence positively sustainable performance in the context of the pharmaceutical industry in the KSA. The resource-based view (RBV) is a management theory that focuses on a firm's internal resources and capabilities as key determinants of its competitive advantage and superior performance. According to the RBV, if a company possesses unique, valuable, and difficult-to-replicate resources, it can achieve sustained competitive advantage over its rivals. This theory suggests that a firm's internal capabilities and resources, such as technology, skilled workforce, and proprietary knowledge, contribute significantly to its long-term success in the marketplace [23].

By applying the RBV, transformational leadership can transform employees and managers into valuable resources and make the human resource management function adopt green practices that help develop, motivate, and provide opportunities to exhibit green behaviours regarding SC practices, as well as achieve a sustainable competitive advantage and superior performance. In parallel with the RBV, the AMO theory is helpful in comprehending how leaders and HR managers can encourage environmentally friendly behaviours and practices in supply chain management. It also explores the connection between HR management and performance, one which suggests that employee skills, motivations, and contributions are essential for sustained performance. This perspective provides an integrated comprehension of how leaders and strategic human resource management managers contribute to organizational performance [26].

As per the AMO theory, HRM practices provide an overarching architecture, influencing (1) employee ability, such as recruitment and selection, training, and development; (2) motivation, such as rewards, incentives, employee empowerment, and compensation; and finally, (3) opportunity, such as teamwork, and empowerment to contribute to the firm's sustainable performance [27]. Through the application of AMO theory, this study expects that pharmaceutical companies' leaders will encourage green HR management and employees' ability, motivation, and opportunity to behave according to environmental management goals and objectives through the green process [28], including the GSCM process.

### 2.2. Green TL and Green HRM

There is a consensus among practitioners and researchers about the vital role of leadership in achieving higher performance. However, the mechanism that links leadership and performance is still unclear and unresolved. In other words, the leadership type can achieve superior performance in different ways [29,30]. GTL could be defined as a style of leadership that inspires and motivates the workforce to meet the organisation's environmental goals while simultaneously supporting their developmental needs [31]. GTL encourages staff members to learn new information and acquire new knowledge [32], and in addition, gets them involved in green process- and product innovation-related activities so that the company can bring green products and services to the market and enhance the company's environmental performance [33]. Leadership is considered a crucial resource in the organisation's green management, including using the RBV as a guide. Transformational executives have a distinct vision for a company's present and future courses of action in the face of competitive marketplaces [34]. As per Zhu et al. [35], transformational leadership encourages better commitment, motivation, trust, and cohesion [35]. The studies of Jia et al. [18] show that intellectually motivated transformational leadership positively affects performance management, talent management, and workforce productivity. GTL embodies the

principals and core values of the top management, which has a significant impact on the company's GHRM [18]. To put it another way, GTL's emphasis on taking each employee's needs into account may motivate them to create and implement GHRM practices to maintain their followers' motivation and empowerment. We anticipate that GTL will support effective GHRM procedures like recruitment and selection, training and learning, performance assessment and management, and remuneration and incentive systems to a greater extent. This inspires, stimulates, and motivates followers to accomplish organisational goals [35]. We suggest that using the AMO theory, GTL uses GHRM to enhance followers' capabilities and skills and furnish opportunities for them to participate in environmental management-related activities for green innovations and ecological performance [36,37]. GTL influences GHRM practices through several key mechanisms: First, green transformational leaders communicate the importance of integrating green practices into HRM strategies, fostering a sense of purpose and alignment among employees in moving towards sustainability goals. They inspire and motivate employees to embrace green initiatives by fostering supportive and empowering work environments. They encourage innovation, creativity, and risk-taking in implementing environmentally friendly HRM practices such as green training programs, performance evaluations, and recruitment strategies. GT leaders can be considered as models in demonstrating their commitment to sustainability through their actions and decisions. Their personal dedication to environmental stewardship sets a positive precedent for employees to follow, influencing the adoption of GHRM practices throughout the organization. Finally, green transformational leaders empower employees to participate actively in green initiatives by providing opportunities for involvement, feedback, and collaboration. They create platforms for employee engagement in decision-making processes related to environmental sustainability, fostering a sense of ownership and responsibility among staff towards green HRM practices [19,30,38,39]. Therefore, we posit that:

**H1a.** *Green transformational leadership positively relates to green human resource management practices.*

### 2.3. Green HRM and Green SCM

GHRM refers to green recruiting and training and the development of green-practices-based performance appraisals, compensation strategies, etc. The adoption of environmental criteria for recruitment, such as personal orientation and environmental competences, are referred to as "green selection and recruitment". The development of green competencies, which enhances organisational performance and capacity and emphasises the importance of the organisation's commitment to green concerns and employee involvement in green processes, is referred to as "green training and involvement", an aspect which gives employees the chance to participate in internal discussions about environmental development. Green performance management and compensation refers to the development of a system for tracking and rewards in order to encourage staff to practice ecological management [40].

Previous research has shown the existence of multiple typologies for Green SCM practices. For instance, Zhu et al. [41] presented a progressive-level-based model of GSCM deployment, starting with basic practices such as green sourcing, and extending to greening the entirety of intra- and inter-organizational processes; this view extends from the suppliers and their contribution, to the product design phase, the transformation phase, and the stage of delivering final products to customers and ensuring reverse logistics. The study emphasised the importance of following GSCM dimensions: intra-organizational environmental practices, green sourcing, green customer relationship management, ecological design, and finally, practices related to the recovery or regeneration of environmental and social capital. This involves making SC investments in eco-friendly or socially responsible practices contributing to long-term well-being and resilience. Prior GSCM empirical investigations have

examined these five dimensions in great detail (e.g., [42–44]). From a structural perspective, Huo et al. [45] presented three groups of GSCM practises, categorised according to the sequence of flows from suppliers to customers. According to the researchers, a firm must integrate green issues into their relationships with customers and suppliers as well as integrate these issues into its internal processes. In the same context, and from a functional perspective, Hervani et al. [46] recommended that green issues should be present in all functions, namely, sourcing, inventory management, production, transportation, distribution, and after-sales services. From a hybrid perspective, Rizki and Augustine [47] counted ten green SC practices: green sourcing, green production, green marketing, green distribution, environmental design, internal environmental management, instruction on the environment, customer relationship management, green information, and green communication technology. Basing their approach on a process view, Zaid et al. [48] adopted two main categories of green SCM practices: (1) internal green SCM, which comprises internal ecological design and management; and (2) external green SCM activities, which include reverse SC, environmental collaboration, and green sourcing. After putting all of the earlier theories up against one another in light of this research, four primary practices are obtained for the current study: (1) green production, maintenance, and inventory management, which refer to internal operations of SCM (INTO); (2) all marketing, distribution, and post-purchase services, which are included in the definition of customer relationship management (CRM); (3) green purchasing, transportation, and other associated processes, which are collectively referred to as supplier relationship management (SRM); and (4) ecological and environmental design (ECOD).

The contribution of GHRM to enhanced GSCM has garnered the interest of scholars and practitioners. The literature has confirmed GHRM's role in acting on and emphasizing green supply chain practices, helping to enhance sustainable performance [48]. GHRM practices impact how successfully the SC and business operate, as claimed by Albahussain et al. [49]. The results of this study demonstrate that human resource practices affect SCP and, consequently, business performance. The results show that SCP may mediate between HRM practices and business performance in Saudi Arabian firms. As per Muafi and Kusumawati [50], GHRM positively affects supply chain organisational learning (SCOL) and SCP. The SCOL has a significant positive effects on SCP and business processes. Furthermore, SC organisational learning mediates the relationship between GHRM and SC performance. Also, SC organisational learning mediates the relationship between GHRM and business processes. The mechanism through which GHRM and GSCM affect financial and environmental performance may be better understood by looking at their relationship. They focus on the mediating roles that effective SCM methods, practices, and outcomes play in the relationship between human resource management and performance. This fact has been concentrated upon by a growing number of conceptual contributions and practical studies in the field of human resource management (e.g., [51,52]). Jabbour and Jabbour [53] have urged empirical research into the HRM–SCM mediation link, particularly as to environmental challenges. Previous empirical studies that looked into this connection revealed that some GHRM practices lead to the adoption of GSCM [54,55]. Actually, in the absence of GHRM practices, employees who are environmentally competent, motivated, and engaged would be lacking, and the adoption of GSCM practices would be hindered by challenges related to classical organisational culture and change management [40,53,56].

Drawing upon RBV theory, it has been demonstrated that GHRM influence positively affects the firm's performance through transforming the staff at all levels into valuable, rare, inimitable, and organised resources that provide the firm with an effective control of intraorganizational and inter-organizational processes within the SC network, and therefore a sustainable competitive advantage [40]. By utilising this viewpoint in the context of environmental management, GHRM may have a vital role in fostering environmental beliefs and values to aid in developing a staff who are environmentally savvy and dedicated and who integrate ecological concepts into SC operations [55,57,58]. Also, Ellinger and

Ellinger [59] and Jabbour and Jabbour [53] emphasised that the GHRM concepts and practices strongly support the implementation of SC practices.

According to [40,48,53,58], GHRM influences GSCM through various key mechanisms as follows: GHRM enhances employee engagement and commitment. In other words, GHRM practices, such as training and development programs focusing on sustainability, environmental awareness, and green initiatives, enhance employees' understanding of and commitment to environmental goals. Engaged and committed employees are more likely to align their actions with green objectives, including those related to the supply chain. Also, GHRM systems incorporate environmental performance metrics and incentives to encourage employees and suppliers to meet sustainability targets. By linking performance evaluations and rewards to green objectives, organizations incentivise behaviours that contribute to environmentally friendly supply chain practices. GHRM provides training and capacity-building opportunities to employees and suppliers on green technologies, practices, and regulations relevant to supply chain operations. By enhancing the skills and knowledge of stakeholders, organizations can improve the implementation of green initiatives across the supply chain. GHRM initiatives drive organizational culture change towards sustainability, influencing attitudes, norms, and behaviours related to environmental stewardship. A culture that values sustainability encourages innovation and adaptation in supply chain processes to minimise environmental impact. GHRM fosters open communication and collaboration among employees, departments, and supply chain partners on sustainability issues. Effective communication channels facilitate the exchange of ideas, best practices, and feedback, enabling continuous improvement in green supply chain management.

Based on the above discussion, it is suggested that the GHRM package of practices positively influences the implementation of GSCM practices. The hypotheses are developed below, and the theoretical model is depicted in Figure 1.

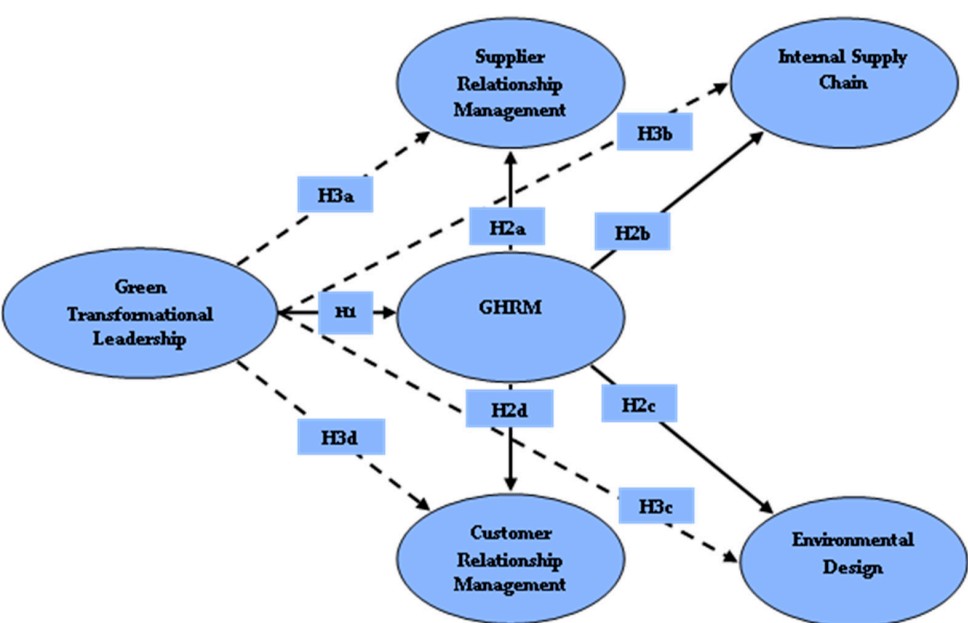

**Figure 1.** Theoretical model.

**H2a.** *GHRM practices are significantly and positively related to SRM.*

**H2b.** *GHRM practices are significantly and positively related to internal SCM.*

**H2c.** *GHRM practices are significantly and positively related to eco-design.*

**H2d.** *GHRM practices are significantly and positively related to CRM.*

In today's organisational dynamics, green transformational leadership (GTL) can significantly shape the environmental orientation of a company. However, realising and implementing green supply chain management (GSCM) practices often involve multifaceted processes. Herein lies the potential mediating role of green human resource management (GHRM), which aligns human resource practices with environmentally conscious values and acts as a conduit that channels and translates visionary green leadership into tangible green practices within the supply chain [38]. By fostering an environmentally aware, motivated, and empowered workforce, GHRM bridges the gap between GTL's vision and the actualisation of sustainable and eco-friendly supply-chain practices. This mediation is crucial in ensuring that the green principles advocated by leadership are seamlessly integrated into the organisational fabric, ultimately contributing to a more sustainable and environmentally responsible business model [39]. Based on this discussion, the hypotheses dealing with the mediation role of GHRM are formulated as follows:

**H3a.** *GHRM significantly and positively moderates the relationship between GTL and SRM.*

**H3b.** *GHRM significantly and positively moderates the relationship between GTL and internal SCM.*

**H3c.** *GHRM significantly and positively moderates the relationship between GTL and eco-design.*

**H3d.** *GHRM significantly and positively moderates the relationship between GTL and CRM.*

From Figure 1 above, in examining the direct impact of GTL on GHRM, it is evident that GTL plays a pivotal role in fostering environmentally conscious practices within organizational human resource management. Through inspirational motivation, idealised influence, individualised consideration, and intellectual stimulation, GTL cultivates a culture of environmental awareness and responsibility among employees. This directly influences the integration of green practices into human resource management processes, such as recruitment, training, performance appraisal, and rewards systems, thereby enhancing overall environmental sustainability efforts within the organization. Furthermore, the indirect effect of GTL on GSCM practices, including SRM, CRM, INTO, and ECOD, mediated through GHRM, underscores the importance of leadership in shaping the broader sustainability initiatives of the organization. By influencing GHRM practices, GTL indirectly impacts the adoption and implementation of green practices throughout the supply chain, facilitating collaborations with suppliers and customers, optimizing internal processes, and designing environmentally friendly products and services. This holistic approach underscores the vital role of leadership in driving sustainable practices across organizational boundaries.

### 3. Research Methodology

*3.1. Research Instrument Design*

The research instrument employed in this study is a comprehensive survey of 28 items designed to measure green transformational leadership, green HRM, and green SCM in the context of pharmaceutical firms (See Appendix A). The primary construct, green transformational leadership, encompasses six items adopted from [18,30,38,39]. The second construct, green HRM, consists of four items adopted from [30,38–40,48,53,58]. The third construct, green SCM, encompasses four key dimensions: (1) supplier relationship management, with five items; (2) internal SC operations, with 5 items; (3) customer relationship management, with four items; and (4) ecological design, with four items. GSCM measures were adopted from [41–43,46–48,53,58]. Respondents are asked to express their agreement or disagreement with each statement using a Likert 5-point scale ranging from 1 (Strongly Disagree) to 5 (Strongly Agree). This instrument is situated within the broader research on green SCM performance, which aims to contribute insights into the nuanced factors influencing the commitment towards green SC practices. Table 1, below, lists the research variables, constructs, and relevant papers used to develop the items for each construct.

**Table 1.** Research instrument description.

| Variables | Constructs | Items | Scales | Related Research |
|-----------|-----------|-------|--------|------------------|
| Green Transformational Leadership | | Six items | | [18,30,38,39] |
| Green HRM | | Four items | | [30,38–40,48,53,58] |
| GSCM | Supplier Relationship Management | Five items | Likert 1–5 | [41–43,46–48,53,58] |
| | Internal SC Operations | Five items | | |
| | Customer Relationship Management | Four items | | |
| | Ecological Design | Four items | | |
| Total | | 28 items | | |

From Table 1 above, GTL encompasses 6 items adopted from [18,30,38,39], GHRM consists of 4 items adopted from [30,38–40,48,53,58], and GSCM, through its four dimensions, consists of 18 items adopted from [41–43,46–48,53,58].

*3.2. Sampling and Data Collection*

This study investigates how GTL influences GSCM practices in Saudi Arabia's pharmaceutical sector. The research will explore the mediating role of green human resource management in understanding the relationship between green transformational leadership and the implementation of environmentally friendly practices within the supply chain. Choosing the right participants is crucial in obtaining precise data for examining the specific connections among all the relevant variables within the research model. This research focuses on managers across the three hierarchical levels within pharmaceutical companies in the Kingdom of Saudi Arabia (KSA). The study encompasses all regions in Saudi Arabia, and the population includes 1221 companies, according to the "Saudi Food and Drug Authority". In any research endeavour, it is crucial to determine the lower limit for sample size. Pacheco et al. [60] observed that achieving a higher level of prediction accuracy necessitates a sample size exceeding 100 responses, which is deemed optimal. In the current study, the G*Power technique was employed [61,62]. The results indicated that the minimum sample size required for this study, which involves six variables, was 89. A total of 111 surveys with comprehensive data were collected. The survey was distributed in both Arabic and English to enhance the response rate. To ensure the authenticity and the high quality of the content of the items, the study undertook a face validity consultation and a pilot investigation. To ensure face validity, the researchers enlisted two seasoned managers employed by pharmaceutical companies and three academic experts specialising in SC and HRM research. The final version of the questionnaire was subsequently formulated based on the insights provided by these experts. The pilot study encompassed 30 responses; the primary results showed that all the items are clear and well formulated.

As for the respondents' backgrounds, the descriptive analysis of the sample reveals vital characteristics and trends among the participants. The sample, comprising 111 respondents from diverse departments within the organisation, exhibits a balanced distribution regarding organisational position, educational level, experience, and gender. In terms of hierarchical level, the majority of respondents hold operational-level positions (41.4%), followed by strategic-level (26%), and tactical-level (23.4%) positions. The descriptive statistics highlight a noteworthy level of educational diversity, ranging from bachelor's degrees (79.3%) to advanced degrees, namely, master's degrees, with 15.3%, and PhD degrees, with 5.4%, contributing to a well-rounded and varied pool of perspectives. In terms of years of experience, the sample indicates in-depth experience in the pharmaceutical field, since

44.1% have less than five years of experience, 26.1% have experience between 5 and 10 years, and 29.8% have more than ten years. In terms of gender representation, the sample is well distributed, with 73% male and 27% female participants. These findings offer a comprehensive snapshot of the sample demographics, providing a solid foundation for further inferential analyses of the variables under investigation.

## 4. Data Analysis

This study employed Partial Least Squares Structural Equation Modelling (PLS-SEM), using SMART-PLS 4 as the statistical method chosen to test its hypotheses. PLS-SEM suits complex research models involving latent variables and multiple relationships. This approach is well-suited for exploratory research and allows for simultaneously assessing both measurement and structural models. PLS-SEM accommodates non-normal data and is robust with smaller sample sizes, making it suitable for this study's context. Through PLS-SEM, the researchers analysed the complex interrelationships between green transformational leadership, green human resource management, and green supply chain practices in the pharmaceutical sector in Saudi Arabia, providing valuable insights into the dynamics of sustainable practices within the industry.

### 4.1. Validity and Reliability Analysis

Validity in research refers to the degree to which an instrument or measure accurately assesses the concept or construct it intends to measure. It is the extent to which a study accurately captures the intended meaning of the variables under investigation. Reliability is the consistency and stability of a measurement tool or instrument over time and across different conditions. A reliable measurement tool produces consistent results, enhancing the credibility and trustworthiness of research findings. As for this research, Figure 2 demonstrates that all items' loadings for all constructs are more than 0.50, which means that all the items are considered when testing hypotheses. Calculations were performed for Alpha Cronbach ($\alpha$), Rho_A, and Composite Reliability (CR) to assess the internal consistency reliability. Table 2 highlights the validity and reliability analyses.

**Table 2.** Validity and reliability results.

| Constructs | $\alpha$ | Rho_A | CR | AVE |
|:---:|:---:|:---:|:---:|:---:|
| CRM | 0.927 | 0.929 | 0.947 | 0.822 |
| INTO | 0.938 | 0.938 | 0.953 | 0.800 |
| ECOD | 0.928 | 0.930 | 0.948 | 0.824 |
| SRM | 0.905 | 0.909 | 0.929 | 0.725 |
| GHRM | 0.964 | 0.964 | 0.974 | 0.902 |
| GTL | 0.959 | 0.960 | 0.967 | 0.830 |

Table 2, above, indicates that all constructs exhibit strong reliability, superior to 0.70. The AVE value, calculated to determine convergent validity, reveals that all values exceed 0.50.

### 4.2. Discriminant Validity

Discriminant validity assesses how much a measurement instrument can distinguish between different constructs or concepts. It examines whether the tool successfully measures distinct, unrelated variables without capturing overlap or redundancy. Cross-load comparisons between the dimensions were conducted to assess discriminant validity and illustrate that the measures of the constructs are not significantly correlated. Table 3 provides outcomes substantiating discriminant validity.

**Table 3.** Discriminant validity.

| CONSTRUCTS | CRM | ECOD | GHRM | GTL | INTO | ARM |
|:---:|:---:|:---:|:---:|:---:|:---:|:---:|
| CRM | 0.906 | | | | | |
| ECOD | 0.857 | 0.907 | | | | |
| GHRM | 0.750 | 0.807 | 0.950 | | | |
| GTL | 0.761 | 0.760 | 0.779 | 0.911 | | |
| INTO | 0.857 | 0.863 | 0.765 | 0.803 | 0.895 | |
| SRM | 0.795 | 0.746 | 0.669 | 0.699 | 0.890 | 0.852 |

The results, in Table 3 above, indicate that for each latent variable (construct), the Average Variance Extracted (AVE) surpasses the construct's highest squared correlation with another latent variable.

*4.3. Hypothesis Testing*

Conducting hypothesis testing comprises the evaluation of two aspects: firstly, scrutinising the path coefficient (β) to understand the overall change in the dependent variable for each alteration in the independent variable; and secondly, employing the t-value test. It is essential to note that irrespective of the (β) value, the t-value must exceed 2 for the coefficient to achieve statistical significance. Table 4 reveals the affirmation of all hypotheses. Specifically, the findings indicate that within pharmaceutical companies in the KSA, green transformational leadership significantly and positively impacts green human resource management (H1). Furthermore, GHRM significantly and positively influences CRM, internal supply chain practices INTO, ECOD, and SRM. In other words, the hypotheses (H2a, H2b, H2c, and H2d) are supported. Furthermore, the results show the positive indirect impact of GTL on CRM, ECOD, INTO, and SRM through the mediating role of GHRM. Consequently, the hypotheses (H3a, H3b, H3c, and H3d) are supported.

**Table 4.** Hypothesis testing.

| Direct Effect | | β | T-Value | P | Decision |
|:---|:---|:---:|:---:|:---:|:---:|
| H1 | GTL ⇒ GHRM | 0.779 | 14.197 | 0.000 | Affirmed |
| H2a | GHRM ⇒ CRM | 0.750 | 18.717 | 0.000 | Affirmed |
| H2b | GHRM ⇒ ECOD | 0.807 | 14.204 | 0.000 | Affirmed |
| H2c | GHRM ⇒ INTO | 0.765 | 11.402 | 0.000 | Affirmed |
| H2d | GHRM ⇒ SRM | 0.660 | 11.015 | 0.000 | Affirmed |
| Indirect Effect | | | | | |
| H3a | GTL ⇒ GHRM ⇒CRM | 0.637 | 7.749 | 0.000 | Affirmed |
| H3b | GTL ⇒ GHRM ⇒ECOD | 0.690 | 8.669 | 0.000 | Affirmed |
| H3c | GTL ⇒ GHRM ⇒INTO | 0.651 | 6.918 | 0.000 | Affirmed |
| H3d | GTL ⇒ GHRM ⇒SRM | 0.572 | 6.179 | 0.000 | Affirmed |

Figure 2 shows the causality relationship between all constructs.

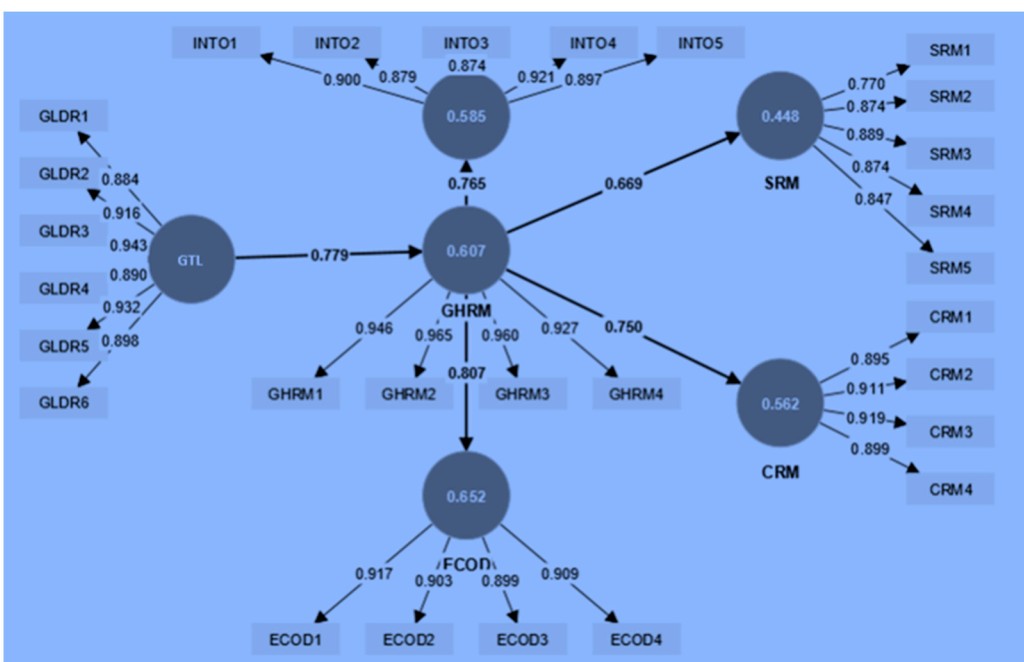

**Figure 2.** SEM model.

Figure 2, depicted above, illustrates that the loadings of all items for each construct exceed 0.50, indicating that all items are eligible for testing the hypotheses. Additionally, the path coefficient (β) demonstrates significance (between 0.669 and 0.807), signifying that all impacts are positively substantial.

## 5. Results Discussion

The findings above indicate a notable level of maturity among pharmaceutical companies operating in the KSA concerning green supply chain management (GSCM) practices and the critical success factors associated with them, particularly green transformational leadership (GTL) and green human resources practices. These pharmaceutical companies are actively working towards meeting the goals outlined in the ambitious Vision 2030, aiming to enhance the health system and attain objectives related to drug security. The results further reveal that H1 is supported, and that GTL significantly influences GHRM practices by supporting and promoting environmentally conscious selection processes, recruitment strategies, performance appraisals, eco-behaviour-based rewards, and training initiatives to acquire new competencies related to sustainable practices. The coefficient of determination indicates that GTL can explain a large percentage of GHRM. These findings align with existing research reported by various scholars, including references such as [18,30,46,60].

The findings also underscore the significance placed on environmental concerns and the regulations reinforcing this movement, which strongly encourage and compel pharmaceutical companies with the aim of their adopting green and sustainable practices. Aligned with the Resource-Based View (RBV) theory, leaders and employees across all hierarchical levels are identified as crucial contributors, providing pharmaceutical companies with the essential capabilities needed to cultivate exemplary green practices, thereby establishing a sustainable competitive advantage. The outcomes about the impact of GHRM practices on GSCM practices show that (H2a, H2b, H2c, and H2d) are supported, and these results align with the conclusions drawn in studies such as [38,48,53,58].

The coefficient of determination ($R^2$) is detailed in Table 5, below. The results indicate that GTL can explain approximately 61% of GHRM. Furthermore, GHRM can explain, respectively, about 56% of CRM, 59% of SRM, 65% of ECOD, and 45% of INTO.

**Table 5.** Coefficient of determination ($R^2$).

| Dependent Variable | $R^2$ |
|---|---|
| Green Human Resource Management | 0.607 |
| Customer Relationship Management | 0.562 |
| Internal Supply Chain | 0.585 |
| Ecological Design | 0.652 |
| Supplier Relationship Management | 0.448 |

## 6. Conclusions

### 6.1. Empirical Results of This Study

The primary aim of this investigation is to explore the correlation between green transformational leadership, in its influence on green human resource management practices, and positive impacts on green supply chain (GSC) practices, a notable challenge faced by pharmaceutical companies in Saudi Arabia (KSA). This study, the first of its kind in the country, seeks to assess the maturity of green supply chain management (GSCM) by scrutinising its foundations, namely green human resource management (GHRM) and green transformational leadership (GTL). Given that the success of Saudi Vision 2030 hinges on the health system and drug security, this research positions itself to contribute by supporting the development of various sectors, including tourism, healthcare, global supply chains, renewable energy, and petrochemicals. Pharmaceutical companies view this vision as an unprecedented opportunity to strengthen their market position and ensure sustainable survival through effective supply chain management (SCM). The study's objective is to gauge the maturity level of GSCM and evaluate the backing provided by GHRM and GTL. Following an extensive literature review, a study model and questionnaire were devised in line with the research methodology. Subsequently, data was collected from a substantial sample, encompassing pharmaceutical businesses in operation within KSA. The analysis emphasises critical factors regarding maturity, like green transformational leadership and green human resource management. The results further underscore the readiness of pharmaceutical firms' executives, managers, and staff to acknowledge the significance of green practices and their potential to enhance the supply chain and, consequently, sustainable performance metrics.

Based on empirical results, Saudi Arabia has emerged as a crucial player in promoting sustainability in the pharmaceutical sector through its commitment to green initiatives, specifically GTL, GHRM, and GSCM. The country's strategic investments in enhancing supply chain resilience, particularly in response to the COVID-19 crisis, and its ambitious sustainability goals outlined in the Vision 2030 agenda, have established a strong foundation for GTL to thrive. Saudi Arabia has created an environment conducive to sustainable practices and organizational change through a focus on environmental stewardship and innovation. Moreover, the country's progressive policies and regulations have encouraged the integration of green principles into human resource management practices, promoting the development of green awareness and a culture of sustainability within organizations. Additionally, Saudi Arabia's strategic location and significant investments in infrastructure have facilitated the implementation of green supply chain management practices, enabling efficient and environmentally responsible sourcing, production, and distribution processes. Through these efforts, Saudi Arabia has become a key contributor to global sustainability initiatives, advancing GTL, GHRM, and GSCM, and positioning itself as a leader in sustainable practices.

*6.2. Research Contribution and Implications*

6.2.1. Theoretical Contribution and Implications

As for the theoretical contributions of this research, this latest effort is a pioneering research project, since it is considered the first of its kind in Saudi Arabia; the study merges three emerging agendas in Saudi Arabia, namely GTL, GHRM, and GSCM, to understand the dynamics of green practices in the Saudi pharmaceutical supply chain. Also, the study contributes to a deeper understanding of the roles played by green transformational leadership and green human resource management in shaping sustainable practices in the supply chain. It highlights the importance of leadership and HRM in driving green initiatives. Second, the development of a research model and questionnaire, informed by a thorough literature analysis, contributes to the methodological aspects of research in this domain. This provides a valuable resource for future studies exploring similar themes or methodologies.

6.2.2. Practical Contribution and Implications

As for practical contribution, the study furnishes leaders and managers of pharmaceutical companies in Saudi Arabia with a comprehensive comprehension of the current state of the diverse factors and enablers aiming at fostering environmentally sustainable practices within the pharmaceutical sector. Also, executives and managers can use the research outcomes to refine their leadership and human resource strategies to impact the green aspects of their supply chain positively. Furthermore, studying the impact of green GTL and GHRM on GSCM practices in the Saudi pharmaceutical sector holds several significant contributions. It provides a unique contextual insight, considering its unique socio-economic, cultural, and regulatory contexts. This enhances the generalizability and applicability of the findings within the Saudi Arabian business environment. The results contribute to an indirect evaluation of the Saudi healthcare system and drug security regarding the Saudi Vision 2030 goals; the study strongly aligns with the national agenda. The results increase managers' awareness of the development of green leadership competencies within organizations, focusing on fostering environmentally conscious decision-making and innovation. This requires aligning HRM strategies in order to recruit, train, and retain employees who prioritise sustainability, while also investing in training and education programs to enhance environmental awareness. Managers would need to revise performance metrics and incentive structures to align with green objectives, collaborate with suppliers to adopt greener practices, ensure regulatory compliance and risk management, stimulate innovation, and engage stakeholders through transparent communication channels. These efforts collectively enable pharmaceutical companies to enhance their environmental performance and create sustainable value for both their businesses and society.

*6.3. Limitations of the Research and Future Research*

In conclusion, this research initiates an examination into the maturity of green transformational leadership (GTL) and its correlation with green human resource management (GHRM) and green supply chain management (GSCM) within the Saudi pharmaceutical industry. Future investigations could extend beyond the current scope by establishing connections between GSC practices and sustainable performance metrics, including environmental, economic, and social dimensions. Given the argument that human resource management (HRM) operationalises the Resource-Based View (RBV) criteria for achieving enduring heightened performance and sustainable competitive advantage, exploring these linkages is crucial. Additionally, the study treated GTL and GHRM as a single construct, but there is merit in scrutinising individual dimensions of GTL and GHRM concerning their impact on GSCM practices. Developing a decision framework for GSCM practices in order to evaluate partnership alternatives with suppliers and customers and their implications on environmental performance could provide further valuable insights.

Moreover, while our focus on the pharmaceutical sector has provided valuable insights, it's important to acknowledge that this narrow scope may limit the generalizability of our findings. Future research endeavours could encompass a broader range of sectors, allowing for a more comprehensive understanding and enabling the extrapolation of results to different industries.

In addition to the above, future research could explore integrating advanced technologies such as blockchain and the Internet of Things (IoT) to enhance traceability, transparency, and overall efficiency. Investigating the impacts and potential benefits of local pharmaceutical manufacturing as to supply chain resilience and a reduced dependency on international suppliers could provide valuable insights. Additionally, research could focus on developing sustainable and environmentally friendly practices within the pharmaceutical supply chain, analysing the adoption of green packaging, energy-efficient transportation, and eco-friendly manufacturing processes. Evaluating the regulatory landscape and compliance with international standards, and exploring collaborative supply chain models and talent management strategies would further contribute to the industry's growth and innovation. The impacts of e-commerce integration and circular economy practices leveraging supplier and customer relationship management (SRM and CRM) in the pharmaceutical supply chain could also be explored to address evolving industry trends and challenges in Saudi Arabia.

**Author Contributions:** Conceptualization, A.L.; methodology, A.L.; software, A.L.; validation, A.L., W.B., B.D. and T.G.; formal analysis, A.L.; investigation, W.B.; resources, B.D.; data curation, T.G.; writing—original draft preparation, A.L.; writing—review and editing, W.B.; visualization, B.D.; supervision, A.L.; project administration, M.A.; funding acquisition, M.A. All authors have read and agreed to the published version of the manuscript.

**Funding:** The authors extend their appreciation to the Deputyship for Research & Innovation, Ministry of Education of Saudi Arabia, for funding this research work through project number (0068-1443-S).

**Institutional Review Board Statement:** Not applicable.

**Informed Consent Statement:** Not applicable.

**Data Availability Statement:** Data is contained within the article.

**Conflicts of Interest:** The authors declare no conflicts of interest.

## Appendix A  Research Instrument

| Variables | Constructs | | Items |
|---|---|---|---|
| Green Transformational Leadership | | 1. | Our leader gives a clear environmental vision for subordinates to follow. |
| | | 2. | Our leader inspires subordinates with environmental plans. |
| | | 3. | The leader makes subordinates work as a team to achieve the same environmental goals. |
| | | 4. | Our leader motivates human resources to reach and achieve environmental goals. |
| | | 5. | Our leader acts in a way that considers the environmental values of individuals. |
| | | 6. | Our leader motivates individuals to present green ideas that take into account the environment. |
| Green Human Resource Management | | 7. | Our company provides appropriate training to promote environmental management as a core organizational value. |
| | | 8. | Our company considers how well an employee is doing at being environmentally friendly as part of their performance evaluations. |
| | | 9. | Our company links employee environmentally friendly behavior to rewards and compensation. |
| | | 10. | When selecting an employee, our company takes into account the environmental dimension in the values and personality of the job applicant. |

| Variables | Constructs | Items |
|---|---|---|
| Green Supply Chain Management | Supplier Relationship Management | 11. Our company's suppliers are obligated to place environmental labels on products.<br>12. Our company receives from suppliers only products that conform to the previously intended specifications and standards.<br>13. Our company cooperates with suppliers to develop sustainable products.<br>14. Our company ensures that the internal management of suppliers adheres to environmental auditing.<br>15. Our suppliers adhere to ISO14000 standards |
| | Internal SC operations | 16. Our company follows a sustainable industrial strategy that takes into account the environment and people.<br>17. There is a commitment in our company to logistics operations that take into account the environmental dimension from senior management and middle management.<br>18. All functions in our company work together to make improvements that respect the environment.<br>19. Our company adheres to comprehensive environmental quality management practices.<br>20. Our company adheres to ISO 14001 certification standards |
| | Customer Relationship Management | 21. The company is keen to know customers' suggestions regarding the design of environmentally friendly medicines<br>22. The company is keen to produce pharmaceutical products that are safe for both society and the environment (cleaner production).<br>23. The company seeks to provide environmentally friendly materials that do not cause negative harm to customers.<br>24. The company cooperates with customers to develop environmentally friendly packaging materials. |
| | Ecological Design | 25. Our company is keen to design products in a way that reduces material/energy consumption.<br>26. Our company is keen to design pharmaceutical products in a way that facilitates easy assembly of raw materials at the lowest possible cost.<br>27. Our company is keen to design products in a way that allows for reuse, recycling and recovery of materials and component parts.<br>28. Our company is keen to design products in a way that allows avoiding hazardous emissions or increased pollution. |
| Total | | 28 items |

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
