# Peer review of "Modelling the Combined Effect of Green Leadership and Human Resource Management in Moving to Green Supply Chain Performance Enhancement in Saudi Arabia"

_sustainability, doi:10.3390/su16103953_

Round 1
Reviewer 1 Report
Comments and Suggestions for Authors
The study has followed the typical steps of the of the research method. The authors should explain all abbreviations. English should be improved.
The authors should explain why the study in KSA is different than in other countries. Managerial implications should be more developped.
Comments on the Quality of English LanguageEnglish should be proofread
Author Response
Dear Reviewer.
I would like to express my sincere gratitude for taking the time to review my research paper. I am committed to incorporating all the recommended changes and addressing the concerns you raised as shown in the table below (The table contains the reviewer’s comments, the correction made by the authors, and the evidence in the manuscript):
|
R1 |
The reviewer’s comments and recommendations |
Corrections made by the authors |
|
1. |
The study has followed the typical steps of the research method. The authors should explain all abbreviations. |
All abbreviations have been explained at the first time and then used in the rest of the manuscript. supply chain management (SCM) supply chains (SC) Kingdom of Saudi Arabia (KSA). Resource-Based View (RBV) Ability-Motivation-Opportunity theory (AMO), internal operations of SCM (INTO); customer relationship management (CRM), supplier relationship management (SRM), ecological and environmental design (ECOD). National Industrial Development and Logistics Program (NIDLP) |
|
2. |
The authors should explain why the study in KSA is different than in other countries. |
A paragraph has been added to show the characteristics of KSA and the challenges that increase the importance of this study….Saudi Arabia's unique context, shaped by its heavy reliance on oil exports and initiatives like Vision 2030, significantly impacts the study and implementation of green leadership, HRM, and SCM. Factors such as the regulatory framework, investment in renewable energy, and water scarcity contribute to specific challenges and opportunities in promoting sustainability within the country……. |
|
3. |
Managerial implications should be more developed. |
Some valuable implications have been added in section 6.2. Research contribution and implications The results increase managers' awareness of the development of green leadership competencies within organizations, focusing on fostering environmentally conscious decision-making and innovation. This requires aligning HRM strategies to recruit, train, and retain employees who prioritize sustainability, while also investing in training and education programs to enhance environmental awareness. Managers would need to revise performance metrics and incentive structures to align with green objectives, collaborate with suppliers to adopt greener practices, ensure regulatory compliance and risk management, stimulate innovation, and engage stakeholders through transparent communication channels. These efforts collectively enable pharmaceutical companies to enhance their environmental performance and create sustainable value for both their businesses and society. |
|
4. |
English should be improved. |
English has been improved using specialized software “Grammarly” |

Reviewer 2 Report
Comments and Suggestions for Authors
I would like to suggest reject for the paper, but I am providing a resubmit recommendation only because you have a survey sample of 111 respondents which could provide some useful insights.
1. The paper is muddled and the aspect of trying to bring together GTL, GHRM, and GSCM requires a lot more work.
2. The research gap is not evident. the discussion about the relationship between GTL and GHRM may seem logical where a Green transformational leader may bring in green human resource policies, but the definitions of either are not strong.
3. Additionally the leap from GHRM to GSCM is quite weak as it pertains to upstream so more on the purchasing side.
4. The hypothesis, specifically between GHRM and the other aspects seems forced. The model is also not logical. It will help if you can define this better and focus more on deriving a strong research gap.
5. The survey instrument is unclear and it will help to publish this in the appendix. Currently it is unclear what you are actually measuring (based on the various references shown for the instrument).
6. The implications are unclear and will become clearer when the papers is rewritten with s stronger research gap and a better model that supports the investigation of the research gap.
7. The survey results and data visualisation of the analysis can be improved.
Comments on the Quality of English Languageoverall Ok, but can be improved when the paper is rewritten with more focus around sentence formation. There are instances of incorrect use of tense.
Author Response
Dear Reviewer.
I would like to express my sincere gratitude for taking the time to review my research paper. I am committed to incorporating all the recommended changes and addressing the concerns you raised, as shown in the table below (The table contains the reviewer’s comments, the correction made by the authors, and the evidence in the manuscript):
|
R2 |
The reviewer’s comments and recommendations |
Corrections made by the authors |
|
1. |
The paper is muddled and the aspect of trying to bring together GTL, GHRM, and GSCM requires a lot more work. |
Many changes have been made to make the relationship among the three variables clear. |
|
2. |
The research gap is not evident. the discussion about the relationship between GTL and GHRM may seem logical where a Green transformational leader may bring in green human resource policies, but the definitions of either are not strong. |
The identified research gap lies in the lack of empirical studies specifically examining the impact of green transformational leadership on green supply chain management (SCM) in Saudi Arabia, considering the mediating role of green human resource management (HRM). While there might be research on each component individually, there seems to be a dearth of comprehensive studies exploring the interconnectedness and potential mediation effects between these three elements within the Saudi Arabian context. This research gap suggests a need for empirical investigations that delve into how green transformational leadership influences green HRM practices, which in turn affect green SCM outcomes in the Saudi Arabian business context. |
|
3. |
Additionally. the leap from GHRM to GSCM is quite weak as it pertains to upstream so more on the purchasing side. The hypothesis, specifically between GHRM and the other aspects seems forced. The model is also not logical. It will help if you can define this better and focus more on deriving a strong research gap. |
Two paragraph have been added to explain the mechanisms through which GTL influences GHRM, and How GHRM practices influence GSCM as well.
GTL can influence GHRM through several mechanism, GTL influences GHRM practices through several key mechanisms: first, green transformational leaders communicate the importance of integrating green practices into HRM strategies, fostering a sense of purpose and alignment among employees towards sustainability goals. They inspire and motivate employees to embrace green initiatives by fostering a supportive and empowering work environment. They encourage innovation, creativity, and risk-taking in implementing environmentally friendly HRM practices, such as green training programs, performance evaluations, and recruitment strategies. GT leaders can be considered as models by demonstrating their commitment to sustainability through their actions and decisions. Their personal dedication to environmental stewardship sets a positive precedent for employees to follow, influencing the adoption of GHRM practices throughout the organization. Finally, green transformational leaders empower employees to participate actively in green initiatives by providing opportunities for involvement, feedback, and collaboration. They create platforms for employee engagement in decision-making processes related to environmental sustainability, fostering a sense of ownership and responsibility among staff towards green HRM practices [19], [30], [38], [39].
GHRM influences GSCM through various key mechanisms as follows: GHRM enhances employee engagement and commitment. In other words, GHRM practices, such as training and development programs focusing on sustainability, environmental awareness, and green initiatives, enhance employees' understanding and commitment to environmental goals. Engaged and committed employees are more likely to align their actions with green objectives, including those related to the supply chain. Also, GHRM systems incorporate environmental performance metrics and incentives to encourage employees and suppliers to meet sustainability targets. By linking performance evaluations and rewards to green objectives, organizations incentivize behaviors that contribute to environmentally friendly supply chain practices. GHRM provides training and capacity-building opportunities to employees and suppliers on green technologies, practices, and regulations relevant to supply chain operations. By enhancing the skills and knowledge of stakeholders, organizations can improve the implementation of green initiatives across the supply chain. GHRM initiatives drive organizational culture change towards sustainability, influencing attitudes, norms, and behaviors related to environmental stewardship. A culture that values sustainability encourages innovation and adaptation in supply chain processes to minimize environmental impact. GHRM fosters open communication and collaboration among employees, departments, and supply chain partners on sustainability issues. Effective communication channels facilitate the exchange of ideas, best practices, and feedback, enabling continuous improvement in green supply chain management. |
|
4. |
The survey instrument is unclear and it will help to publish this in the appendix. Currently it is unclear what you are actually measuring (based on the various references shown for the instrument). |
The questionnaire has been added as Appendix A before the references. (Appendix A). |
|
5. |
The implications are unclear and will become clearer when the papers is rewritten with s stronger research gap and a better model that supports the investigation of the research gap. |
Practical implications have been strengthened by adding more ones. The results increase managers’ awareness of the development of green leadership competencies within organizations, focusing on fostering environmentally conscious decision-making and innovation. This requires aligning HRM strategies to recruit, train, and retain employees who prioritize sustainability, while also investing in training and education programs to enhance environmental awareness. Managers would need to revise performance metrics and incentive structures to align with green objectives, collaborate with suppliers to adopt greener practices, ensure regulatory compliance and risk management, stimulate innovation, and engage stakeholders through transparent communication channels. These efforts collectively enable pharmaceutical companies to enhance their environmental performance and create sustainable value for both their businesses and society. |
|
6. |
The survey results and data visualisation of the analysis can be improved. |
A paragraph under each figure or table have been added. Also, some information such as the software used, questionnaire have been added. |

Reviewer 3 Report
Comments and Suggestions for Authors
First part of Abstract is actually Introduction - lines 1-15.
In the Abstract there should be clearly written research question or hypothesis. Now is missing.
Missing info about the research method in the Abstract.
Missing research gap in the Introduction. Should be added. f.e. after line 121.
No research problem is mentioned in the Article. Should be added. f.e after line 133.
Line 175 - (GTL) - should be added.
After Figure 1- I suggest comment, explanation of the figure in the text, before Chapter 3.
After Table 1- I suggest comments, explanations.
Line 363- 1221 companies - SFDA - meaning should be explained.
Chapter 4. which software was used, version? - should be added.
After Table 2- I suggest comments, explanations.
After Table 3 - I suggest comments, explanations.
After Figure 1 line 474 - I suggest comments, explanations.
Figure 1 is already on the page 7. Here on the page 11 should be Figure 2.
After Table 5 line 507 - I suggest comments, explanations.
Line 508 and chapter 6.1 there should be clearly written for each hypothesis whether is confirmed or no in the text.
Implications - separate please practical and theoretical ones.
Author Response
Dear Reviewer.
I would like to express my sincere gratitude for taking the time to review my research paper. I am committed to incorporating all the recommended changes and addressing the concerns you raised as shown in the table below (The table contains the reviewer’s comments, the correction made by the authors, and the evidence in the manuscript):
|
R3 |
The reviewer’s comments and recommendations |
Corrections made by the authors |
|
1. |
First part of Abstract is actually Introduction - lines 1-15. |
Lines 1-15 have been rephrased to put more focus on the issue dealt with in the manuscript…..Previous research was limited in examining the causal relationship between Green transforma-tional leadership (GTL) and green supply chain management (GSCM) with the intermediary in-fluence of Green human resource management (GHRM) within the pharmaceutical sector of the Kingdom of Saudi Arabia (KSA). This gap persisted despite the recognition in Saudi Vision 2030 of logistics and supply chain management (SCM) as fundamental to the national development agenda, given that contemporary competitiveness lies in the efficacy of supply chains (SC) rather than individual companies. Moreover, the achievement of economic progress hinges significantly on how well these accomplishments align with sustainability demands and obstacles…… |
|
2. |
In the Abstract there should be clearly written research question or hypothesis. Now is missing. |
The research question has been added to the introduction. This paper aims to investigate the extent to which GTL foster GRHM practices to enhance the maturity of GSCM performance in the pharmaceutical industry in KSA?. In other words, the research goal is to explain the variance of GSCM due to GHRM and GTL. |
|
3. |
Missing info about the research method in the Abstract. |
Information about the methodology has been added….The methodology employed is deductive and translated into a questionnaire derived from a comprehensive review of existing literature. This questionnaire was subsequently tested through the collection of 111 responses from pharmaceutical companies operating in KSA. |
|
4. |
Missing research gap in the Introduction. Should be added. f.e. after line 121. |
The research gap has been added in a clear way as follows: The identified research gap lies in the lack of empirical studies specifically examining the impact of green transformational leadership on green supply chain management (SCM) in Saudi Arabia, considering the mediating role of green human resource management (HRM). While there might be research on each component individually, there seems to be a dearth of comprehensive studies exploring the interconnectedness and potential mediation effects between these three elements within the Saudi Arabian context. This research gap suggests a need for empirical investigations that delve into how green transformational leadership influences green HRM practices, which in turn affect green SCM outcomes in the Saudi Arabian business context. |
|
5. |
No research problem is mentioned in the Article. Should be added. f.e after line 133. |
The research problem that refers to the issue and the gap of knowledge addressed in this manuscript has been merged with the research gap paragraph. The identified research gap lies in the lack of empirical studies specifically examining the impact of green transformational leadership on green supply chain management (SCM) in Saudi Arabia, considering the mediating role of green human resource management (HRM). While there might be research on each component individually, there seems to be a dearth of comprehensive studies exploring the interconnectedness and potential mediation effects between these three elements within the Saudi Arabian context. This research gap suggests a need for empirical investigations that delve into how green transformational leadership influences green HRM practices, which in turn affect green SCM outcomes in the Saudi Arabian business context. |
|
6. |
Line 175 - (GTL) - should be added. |
The abbreviation has been added……GTL could be defined as a style…… |
|
7. |
After Figure 1- I suggest comment, explanation of the figure in the text, before Chapter 3. |
A paragraph has been added after figure 1. From figure 1 above, in examining the direct impact of GTL on GHRM, it is evident that GTL plays a pivotal role in fostering environmentally conscious practices within organizational human resource management. Through inspirational motivation, idealized influence, individualized consideration, and intellectual stimulation, GTL cultivates a culture of environmental awareness and responsibility among employees. This directly influences the integration of green practices into human resource management processes, such as recruitment, training, performance appraisal, and rewards systems, thereby enhancing overall environmental sustainability efforts within the organization. Furthermore, the indirect effect of GTL on GSCM practices, including SRM, CRM, INTO, and ECOD, mediated through GHRM, underscores the importance of leadership in shaping the broader sustainability initiatives of the organization. By influencing GHRM practices, GTL indirectly impacts the adoption and implementation of green practices throughout the supply chain, facilitating collaboration with suppliers and customers, optimizing internal processes, and designing environmentally friendly products and services. This holistic approach underscores the vital role of leadership in driving sustainable practices across organizational boundaries.
|
|
8. |
After Table 1- I suggest comments, explanations. |
A paragraph has been added after Table 1. From Table 1 above, GTL englobes 6 items adopted from [19], [30], [58], [59], GHRM consists of 4 items adopted from [30], [38], [47], [52], [57], [58], [59], GSCM, through its four dimensions, consists of 18 items adopted from [39], [40], [42], [43], [45], [46], [47], [52], [57].
|
|
9. |
Line 363- 1221 companies - SFDA - meaning should be explained. |
The acronym has been explained…..according to “Saudi Food and Drug Authority”. |
|
10. |
Chapter 4. which software was used, version? - should be added. |
The software and version has been added…..This study employed Partial Least Squares Structural Equation Modeling (PLS-SEM) using SMART-PLS 4 as the chosen statistical method to test its hypotheses |
|
11. |
After Table 2- I suggest comments, explanations. |
A paragraph has been added after Table 2. Table 2 above indicates that all constructs exhibit strong reliability, superior to 0.70. The AVE value, calculated to determine convergent validity, reveals that all values exceed 0.50.
|
|
12. |
After Table 3 - I suggest comments, explanations. |
A paragraph has been added after Table 3. The results, in Table 3 above, indicate that for each latent variable (construct), the Average Variance Extracted (AVE) surpasses the construct's highest squared correlation with another latent variable.
|
|
13. |
After Figure 2 line 474 - I suggest comments, explanations. |
A paragraph has been added after Figure 2. Figure 2 depicted above illustrates that the loadings of all items for each construct exceed 0.50, indicating that all items are eligible for testing the hypotheses. Additionally, the path coefficient (β) demonstrates significance (between 0.669 and 0.807), signifying that all impacts are positively substantial.
|
|
14. |
Figure 1 is already on the page 7. Here on the page 11 should be Figure 2. |
Figure 2 shows the results related to each hypothesis in Figure 1. |
|
15. |
After Table 5 line 507 - I suggest comments, explanations. |
A paragraph has been added after Table 5. The coefficient of determination (R2) is detailed in Table 5 above. The results indicate that GTL can explain approximately 61% of GHRM. Furthermore, GHRM explain respectively about 56% of CRM, 59% of SRM, 65% of ECOD, and 45% of INTO. |
|
16. |
Line 508 and chapter 6.1 there should be clearly written for each hypothesis whether is confirmed or no in the text. |
The judgement for each variable has been clearly mentioned in hypothesis testing and the discussion sections…. Specifically, the findings indicate that within pharmaceutical companies in KSA, green transformational leadership significantly and positively impacts green human resource management (H1). GHRM significantly and positively influences CRM, internal supply chain practices INTO, ECOD, and SRM. In other words, the hypotheses (H2a, H2b, H2c, H2d) are supported.
the positive indirect impact of GTL on CRM, ECOD, INTO and SRM through the mediation role of GHRM. Consequently, the hypotheses (H3a, H3b, H3c, H3d) are supported.
that H1 is supported and that GTL significantly influences GHRM practices by supporting and promoting environmentally conscious selection processes, recruitment strategies, performance appraisals, eco-behaviour-based rewards, and training initiatives to acquire new competencies related to sustainable practices. The coefficient of determination indicates that GTL can explain approximately 61% of GHRM. These findings align with existing research reported by various scholars, including references such as [19], [30], [58] and [59]. The outcomes about the impact of green human resource management practices on Green Supply Chain (GSC) practices show that (H2a, H2b, H2c, H2d) are supported and these results align with the conclusions drawn in studies such as [38], [47], [52] and [57]. |
|
17. |
Implications - separate please practical and theoretical ones. |
The separation is done as follows: 6.2. Research contribution and implications 6.2.1. Theoretical contribution and implications 6.2.2. Practical contribution and implications |

Round 2
Reviewer 2 Report
Comments and Suggestions for Authors
1. The research is focused on KSA not as a research gap rather your sample is from KSA. There should be clarity in terms of what exactly is the research gap and why it is important within the KSA context as a gap (excluding the 2030 vision)
2. The sample is pharma so cannot be generalised as it is context specific.
3. The sample of 111 is low in terms of the variables, and also confusing in terms of the three hierarchical levels. How do you correlate between the hierarchical levels and leadership perspective. Also, are all respondents focused on suppliers/ customers/ product design? Where do they get the perspectives from?
4. The model has discrete dependent variables, although SRM and IntO are clubbed under GSCM, but there is no correlation between these.
5. Is SCM required? The measures of SRM are only focused on green initiatives, not on SRM
6. The model needs more thought. You could remove GSCM out of this and only focus on product design and customer based on the sample population.
7. It is not clear how the questionnaire was designed. You have mentioned that there was a pilot study, did anything change after the study? Did you use variables of TL within a green setting or did you use any specific Green TL variable? This is not clear. The leap from GTL to GHRM to other variables seems Ok from a calculation perspective, but from the aspect of the sample population of three hierarchies and perhaps not associated with the functions, it is unclear.
Author Response
Dear Reviewer.
I sincerely appreciate your thoughtful review and insightful comments. Your feedback has truly broadened my perspective and highlighted several important areas that warrant careful consideration. Thank you for your valuable input; it has been instrumental in enhancing the quality and depth of my work. I am committed to incorporating all the recommended changes and addressing the concerns you raised, as shown in the table below (The table contains the reviewer’s comments, the correction made by the authors, and the evidence in the manuscript):
|
R2 |
The reviewer’s comments and recommendations |
Corrections made by the authors |
|
1. |
1. The research is focused on KSA not as a research gap rather your sample is from KSA. There should be clarity in terms of what exactly is the research gap and why it is important within the KSA context as a gap (excluding the 2030 vision) |
I have endeavored to fulfill your specifications for the manuscript correction to the best of my ability. I changed significantly the manuscript from the first version. I do really hope I can satisfy you requirements.
I do really agree with your opinion that the discussion requires more clarifications to link the results and discussion with Saudi context….for this, some clarifications have been added as follows: Based on empirical results, Saudi Arabia has emerged as a crucial player in promoting sustainability in the pharmaceutical sector through its commitment to green initiatives, specifically GTL, GHRM, and GSCM. The country's strategic investments in enhancing supply chain resilience, particularly in response to the Covid-19 crisis, and its ambitious sustainability goals outlined in the Vision 2030 agenda have established a strong foundation for GTL to thrive. Saudi Arabia has created an environment conducive to sustainable practices and organizational change through a focus on environmental stewardship and innovation. Moreover, the country's progressive policies and regulations have encouraged the integration of green principles into human resource management practices, promoting the development of green awareness and a culture of sustainability within organizations. Additionally, Saudi Arabia's strategic location and significant investments in infrastructure have facilitated the implementation of green supply chain management practices, enabling efficient and environmentally responsible sourcing, production, and distribution processes. Through these efforts, Saudi Arabia has become a key contributor to global sustainability initiatives, advancing GTL, GHRM, and GSCM, and positioning itself as a leader in sustainable practices.
|
|
2. |
2. The sample is pharma so cannot be generalised as it is context specific. |
I do really agree with your point of view. This issue has been mentioned as a limitation as follows: Moreover, while our focus on the pharmaceutical sector has provided valuable insights, it's important to acknowledge that this narrow scope may limit the generalizability of our findings. Future research endeavors could encompass a broader range of sectors, allowing for a more comprehensive understanding and enabling the extrapolation of results to different industries. |
|
3. |
3. The sample of 111 is low in terms of the variables, and also confusing in terms of the three hierarchical levels. How do you correlate between the hierarchical levels and leadership perspective. Also, are all respondents focused on suppliers/ customers/ product design? Where do they get the perspectives from? |
I agree that the larger the sample size, the more reliable the results. It is true that we didn’t achieve the maximum sample size, but the minimum is surpassed based on G*Power technique (89 responses). An additional paragraph has been added to explain this as follows: In any research endeavor, it is crucial to determine the lower limit for sample size. Pacheco et al. [59] observed that achieving a higher level of prediction accuracy necessitates a sample size exceeding 100 responses, which is deemed optimal. In the current study, the G*Power technique was employed [60], [61], [62]. The results indicated that the minimum sample size required for this study, which involves six variables, was 89.
Also, SmartPLS is often considered useful for small sample sizes because it employs a partial least squares (PLS) approach, which is particularly suited for situations with limited data. Unlike covariance-based structural equation modeling (CB-SEM), PLS does not have strict requirements for sample size, making it more flexible and robust for analyzing smaller datasets.
|
|
4. |
4. The model has discrete dependent variables, although SRM and IntO are clubbed under GSCM, but there is no correlation between these. |
Numerous scholars, including Chopra & Meindl (2013) and Christopher (2005), have noted that Supplier Relationship Management (SRM) and Customer Relationship Management (CRM) can function as dependent variables for Internal Supply Chain Practices (INTO). This is because the success of SRM and CRM practices hinges on effective collaboration within the organization across functions, as well as the implementation of quality standards such as ISO 14001. Since this research aims to explore the mediating role of Green Human Resource Management (GHRM) between Green Transformational Leadership (GTL) and Green Supply Chain Management (GSCM) practices, the study did not consider the causal relationships among GSCM practices. |
|
5. |
5. Is SCM required? The measures of SRM are only focused on green initiatives, not on SRM |
It is important to adopt appropriate measures for all constructs. Our research endeavored to do this through adopting the measures from validated research as shown in table 1 ( [41], [42], [44], [45], [46], [47], [52], [57], [58]). Also, initiatives and practices are interdependent. Initiatives set the direction and goals, while practices are the means by which these goals are achieved. Effective initiatives require appropriate practices to be successful, and successful practices often emerge from well-designed initiatives. Based on your valuable comment, investigating new typologies for future research seems to be very important. |
|
6. |
6. The model needs more thought. You could remove GSCM out of this and only focus on product design and customer based on the sample population. |
Your comment significantly gives many perspectives to broaden the horizons for future research as now it is already done. |
|
7. |
7. It is not clear how the questionnaire was designed. You have mentioned that there was a pilot study, did anything change after the study? Did you use variables of TL within a green setting or did you use any specific Green TL variable? This is not clear. The leap from GTL to GHRM to other variables seems Ok from a calculation perspective, but from the aspect of the sample population of three hierarchies and perhaps not associated with the functions, it is unclear. |
More clarifications about the questionnaire development have been added…..The primary construct, green transformational leadership, encompasses six items adopted from [19], [30], [38-39]. The second construct, green HRM, consists of four items adopted from [30], [38-40], [47], [52], [57] . The third construct, green SCM, encompasses four key dimensions: (1) Supplier Relationship Management, with five items. (2) Internal SC operations with 5 items. (3) Customer Relationship Management with four items. (4) ecological design with four items. GSCM measures were adopted from [41-42], [44-47], [52], [57-58].
The paragraph has been improved to show that the research has undertaken face validity from experts and a pilot study of 30 responses.
To ensure the authenticity and high-quality content of the items, the study undertakes a face validity and a pilot investigation. To ensure face validity, the researchers enlisted two seasoned managers employed in pharmaceutical companies and three academic experts specialising in SC and HRM research. The final version of the questionnaire was subsequently formulated based on the insights provided by these experts. The pilot study has encompassed 30 responses, the primary results showed that all the items are cleat and well formalted.
For GTL, as shown in Table 1, previous research used to develop the GTL measures were specific GTL and not TL in a green setting. Here are some examples of the research used:
· S. K. Singh, M. Del Giudice, R. Chierici, and D. Graziano, “Green innovation and environmental performance: The role of green transformational leadership and green human resource management,” Technological forecasting and social change, vol. 150, p. 119762, 2020. · T. Chen and Z. Wu, “How to facilitate employees’ green behavior? The joint role of green human resource management practice and green transformational leadership,” Frontiers in psychology, vol. 13, p. 906869, 2022. · J. Peng, K. Yin, N. Hou, Y. Zou, and Q. Nie, “How to facilitate employee green behavior: The joint role of green transformational leadership and green human resource management practice,” Acta Psychologica Sinica, vol. 52, no. 9, p. 1105, 2020.
The questionnaires were distributed to employees occupying either top-level positions, providing them with a holistic, cross-functional perspective on the company's operations, or those involved in specific logistics functions. |

Round 3
Reviewer 2 Report
Comments and Suggestions for Authors
Dear Author/s
Thank you for the changes to the paper. Although I think there is more scope for improvement, I will recommend acceptance.